# Neoantigen Vaccines; Clinical Trials, Classes, Indications, Adjuvants and Combinatorial Treatments

**DOI:** 10.3390/cancers14205163

**Published:** 2022-10-21

**Authors:** Jenni Viivi Linnea Niemi, Aleksandr V. Sokolov, Helgi B. Schiöth

**Affiliations:** Department of Surgical Sciences, Functional Pharmacology and Neuroscience, Uppsala University, 751 24 Uppsala, Sweden

**Keywords:** neoantigen vaccines, cancer therapy, cancer vaccine, personalized therapy

## Abstract

**Simple Summary:**

Personalized neoantigen vaccines are a diverse group of personally tailored cancer vaccines that strengthen patients´ own immune reaction against cancer antigens. We analyzed 147 neoantigen vaccine clinical trials from ClinicalTrials.gov database and showed that peptide vaccines were the dominating vaccine type, while there were multiple new neoantigen vaccine types in the field. We also showed that neoantigen vaccines were mostly used in the treatment of small cell lung cancer, non-small cell lung cancer, melanoma, and glioma. According to our results neoantigen vaccines work at their best when combined with other treatments such as immune checkpoint inhibitors, chemotherapy, and radiation therapy. The effect of some neoantigen vaccine types were also often promoted with adjuvant therapy where poly-ICLC were the most recurrent adjuvant choice.

**Abstract:**

Personalized neoantigen vaccines are a highly specific cancer treatment designed to induce a robust cytotoxic T-cell attack against a patient’s cancer antigens. In this study, we searched ClinicalTrials.gov for neoantigen vaccine clinical trials and systematically analyzed them, a total of 147 trials. Peptide vaccines are the largest neoantigen vaccine type, comprising up to 41% of the clinical trials. However, mRNA vaccines are a growing neoantigen vaccine group, especially in the most recent clinical trials. The most common cancer types in the clinical trials are glioma, lung cancer, and malignant melanoma, being seen in more than half of the clinical trials. Small-cell lung cancer and non-small-cell lung cancer are the largest individual cancer types. According to the results from the clinical trials, neoantigen vaccines work best when combined with other cancer treatments, and popular combination treatments include immune checkpoint inhibitors, chemotherapy, and radiation therapy. Additionally, half of the clinical trials combined neoantigen vaccines with an adjuvant to boost the immune effects, with poly-ICLC being the most recurrent adjuvant choice. This study clarifies the rapid clinical trial development of personalized neoantigen vaccines as an emerging class of cancer treatment with increasingly diversified opportunities in classes, indications, and combinatorial treatments.

## 1. Introduction

Immunotherapy has revolutionized cancer treatments by minimizing adverse events, attacking the cancer microenvironment, and increasing the overall patient survival rate [1]. The aim of immunotherapy is to boost patients´ own T-cell immunity to recognize and eradicate the tumor [2,3,4,5]. Cancer vaccines are an attractive alternative immunotherapy option, where antigens from the tumor are selected and injected back into the patient to induce a highly targeted antitumor immune reaction [4,6].

Early therapeutic vaccinations in this field were vaccines based on tumor-associated antigens (TAAs): self-antigens that are overexpressed by tumor cells [3,6,7,8]. TAAs have met with limited success due to low tumor specificity as TAAs are not unique to the tumor cells. This leads to a high risk of off-target toxicity and autoimmunity [3,7,8]. Furthermore, TAAs elicit poor inoculation efficiency as the immune system is hard-wired not to attack factors expressed by our own cells by mechanisms of central and peripheral tolerance [3,9,10].

A solution to this problem is *personalized* neoantigen vaccines, which are based on tumor-specific antigens generated by non-synonymous mutations in the tumor, and thus exclusively expressed in tumor cells [7,9]. This provides an option for tumor destruction without hurting healthy tissue and provides the possibility for long-term protection against tumor recurrence through post-treatment immunological memory [7,11]. Personalized neoantigen vaccines have provided promising data showing the immune system is effectively configurated to perform the dirty work of killing cancer [3,4,7].

Recent review articles have provided important information on the advances in neoantigen vaccine immunotherapy. However, most current articles take a narrative perspective on the topic and only a few review articles systematically analyze the present clinical data. Esprit et al. reviewed studies with mRNA-based neoantigen vaccines, analyzing 13 clinical trials. According to this article, evaluating immune activity in the early stages of a clinical trial could allow for improved assessment of neoantigen response. This could reveal a minimal immunogenicity score that a neoantigen vaccine should adhere to [4]. Chen et al. analyzed clinical trials involving neoantigen vaccines in the treatment of pancreatic ductal adenocarcinoma, and they point out that neoantigen immunotherapy is currently hindered by specialized tumor microenvironments, complex treatment protocols, and insufficient antigenicity [8]. Mattos-Arruda et al., on the other hand, analyzed 15 selected clinical trials targeting neoantigens. The article put forward a hypothesis that personalized neoantigen vaccines will help to solve three major challenges regarding cancer immunotherapy: tumor heterogeneity, the diversity of malignant clones per patient, and the severe side effects [11,12]. According to recent review articles, Hodge et al. had the most comprehensive systematical analyses of neoantigen vaccines in the treatment of different cancer types [3].

The previous articles include valuable systematic analysis of the clinical trials, but the field is developing rapidly, and there is a need for analysis that compares neoantigen vaccine clinical trials. To the best of our knowledge, there are no published review articles about neoantigen vaccine clinical trials that would extensively cover the whole field. Therefore, we chose to include all the available neoantigen vaccine clinical trials and quantitatively evaluate their features, such as status, results, and indications. Here, we provide a detailed analysis of neoantigen vaccines in cancer treatment from the first trial in 1997 to today. We include neoantigen vaccine types that are commonly not included as their own group and show how trends in the popularity of different vaccine types develop over time. We also show how cancer types vary for different indications and explore potential reasons for the popularity of certain vaccines. Furthermore, we illustrate the usage of adjuvant therapies and combinatorial treatments. The aim is to provide a quantitative perspective and illustrate the diversity of classes, indications, and combinations in current clinical trials and discuss the roles of neoantigen vaccines for cancer treatment.

## 2. Materials and Methods

The clinical trials were downloaded from the database ClinicalTrials.gov. The methodological approach was similar to that described in our previous studies [13]. Search results were gathered using the search terms “Cancer Vaccine” and “Neoantigen Vaccines”. We included all the studies using neoantigen vaccines as a cancer treatment with all start dates, age groups, and cancer types. The search yielded an initial 1316 clinical trials. These clinical trials were manually assessed to determine if they really had neoantigen vaccines as a cancer treatment. Most of the clinical trials had some other type of cancer vaccine as a treatment, and all these trials were excluded from the data. A total of 147 clinical trials were selected after the manual selection. To obtain data about the vaccine type, cancer type, adjuvant, and combinatorial treatments, we manually analyzed trial arm descriptions, interventions, and trial descriptions. The control arms were not included in the quantitative analysis. All these clinical trials can be seen in Appendix A.

In this analysis, the status for clinical trials was divided into six categories: Not yet active, Completed, Active, Not recruiting, Terminated, Unknown status, and Withdrawn. Neoantigen vaccines were divided into 12 different categories instead of the 4 in which they are commonly divided. These additional categories are included to provide a better picture of the different neoantigen vaccines in the current clinical trials. The information was purely collected according to the vaccine type given in the article. Cancer types were mainly divided according to the location of cancer. Lung cancer, small-cell lung cancer, and non-small-cell lung cancer are specified as their own groups, and for glioma, glioblastoma is separated as its own group. Additionally, pediatric brain tumors may also include glioma and glioblastoma in children, but these are not specified in these clinical trials and are therefore included as separate groups.

## 3. Results

The first neoantigen vaccine trial was initiated in 1997, and the next ones in 1999, 2001, and 2004, after which the number of clinical trials increased stably (Figure 1A). The number of clinical trials reached a peak in 2018 and 2019, and then slightly decreased. Of the clinical trials, 46% were classified as Not yet active, 19% were Completed, and the rest of them were Active but not recruiting, Terminated, Withdrawn, or had an unknown status (Figure 1B).

## 4. Vaccine Types

### 4.1. Peptide Vaccines

The clear majority of the clinical trials used peptide neoantigen vaccines (Figure 1C). Peptide vaccines are generally divided into long peptides (15–31 amino acids in length) and short peptides (8–10 amino acids in length). Short peptides are commonly used in earlier clinical trials, while newer clinical trials use long-peptide neoantigen vaccines which induce both CD4+ and CD8+ T-cell responses to overcome immune tolerance [3].

Problems with neoantigen peptide vaccines are related to a limited immune response that is often weak and temporary. In order to avoid this problem, poly-ICLC adjuvant is commonly used with peptide vaccines to mimic dsRNA to activate TLR3 [3]. Additionally, the peptide vaccine generation time is generally 2–3 months and can be too long for many patients with a limited life expectancy [14].

NeoVax was the most popular peptide vaccine and is often combined with a poly-ICLC adjuvant. It is commonly administrated subcutaneously and can be combined with other treatments, such as nivolumab. For instance, the clinical trial NCT03219450 used NeoVax for the first time in combination with cyclophosphamide and pembrolizumab for patients suffering from chronic lymphocytic leukemia. Another NeoVax trial tests vaccination and poly-ICLC as monotherapy (in addition to surgery) against melanoma. This study aims to establish if a patient’s immune response is able to induce a strong reaction against the tumor cells with the help of the vaccine (NCT01970358). NCT04943718, on the other hand, tests a peptide-type neoantigen vaccine as monotherapy after surgery, radiotherapy, and chemotherapy for recurrent malignant glioma patients. Interestingly, this trial does not include an adjuvant, and one of the inclusion criteria is that patients must not have received any immune therapy.

Other interesting peptide vaccine clinical trials include NCT03012100, which tests the multi-epitope folate receptor alpha peptide vaccine combined with sargramostim (GM-CSF) and cyclophosphamide chemotherapy in treating triple-negative breast cancer. These vaccines are created from a person´s white blood cells, which are mixed with tumor proteins. In this case, the vaccination is aimed at preventing disease relapse after surgery and other standard treatments (NCT03012100). This illustrates the variability in peptide vaccine treatment aims, as many neoantigen vaccine clinical trials use peptide vaccines to directly attack uncontrollably growing tumors instead of preventing relapse. In the clinical trial NCT02600949, for instance, peptide vaccines are trialed for treatment of spreading pancreatic or colorectal cancers that cannot normally be cured or controlled with available treatments. In this clinical trial, a peptide vaccine is combined with imiquimod, pembrolizumab, and sotigalimab (NCT02600949).

### 4.2. Heat Shock Protein Vaccines

Heat shock protein vaccines are the group of neoantigen vaccines that has the highest number of reported results at the moment. The HSPPC-96 vaccine, for instance, has been used in combination with temozolomide in the treatment of glioblastoma multiforme. The median progression-free survival was observed to be 18 months with adverse events such as anemia, fatigue, and hyperglycemia (NCT00905060). In another study, the same HSPPC-96 vaccine was studied without checkpoint inhibitory therapy. In this study, the median progression-free survival was considerably lower (19.1 weeks), whereas the median overall survival was 42.6 weeks. Focal deficit, fatigue, and gastrointestinal disorders were some of the observed adverse events in this study (NCT00293423).

The HSPPC-96 vaccine has also been tested in combination with bevacizumab. Importantly, this clinical trial reported results that show that patients that only received bevacizumab without HSPPC-96 actually had the highest overall survival (10.7 months) and progression-free survival (5.3 months) compared to the patients that received both bevacizumab and the HSPPC-96 vaccine (NCT01814813). These results are important as they illustrate heat shock protein vaccines´ negative treatment outcome and show that heat shock protein vaccines are not only associated with positive results. This study did not include HSPPC-96 vaccine treatment alone and therefore, the overall survival for patients administered the HSPPC-96 vaccine alone cannot be compared to those given bevacizumab treatment alone. The outcome of this study potentially indicates that bevacizumab and HSPPC-96 should not be combined as a treatment.

Other heat shock protein vaccine types include gp96, which is trialed as a post-operation treatment for liver cancer patients in NCT04206254 or rHSC-DIPGVax and is mainly used for childhood diffuse intrinsic pontine gliomas or diffuse midline gliomas (NCT04206254) (NCT04943848).

### 4.3. Dendritic Cell Vaccines

The second most popular neoantigen vaccine candidates are dendritic cell-based vaccines. There are multiple dendritic cell subsets available for vaccination, with monocyte-derived dendritic cells and leukemia-derived dendritic cells being the two main types. Dendritic cell vaccines can be used to promote antitumor immunity by using different strategies, such as in situ vaccination and in vitro canonical vaccination. In situ vaccination promotes dendritic cell uptake and tumor recognition by releasing tumor antigens locally through standard therapies, whereas canonical vaccination loads tumor antigens into dendritic cells before delivering dendritic cells to patients [15].

One study with reported results compared the efficacy of ICT-107 dendritic cells pulsed with immunogenic peptides from tumor antigens with the efficacy of dendritic cells that had not been pulsed with antigens. The results show that dendritic cells pulsed with antigens were observed to have a median overall survival of 18.3 months, whereas dendritic cells not pulsed with antigens had an overall survival of only 16.7 months. In a similar fashion, the progression-free survival was also higher for the dendritic cells pulsed with antigens than for the dendritic cells not pulsed with antigens. Therefore, according to the results of this trial, dendritic cells with pulsed antigens would be a more effective treatment alternative (NCT01280552). The study NCT02332889 also reported results, but in this trial, the results are less encouraging. A patient received autologous dendritic vaccines for treatment of relapsed pediatric high-grade glioma. The vaccine was combined with poly-ICLC and decitabine, but it failed to halt disease progression (NCT02332889). The study NCT01067287, on the other hand, trialed dendritic cell vaccines with anti-PD1 blockage following stem cell transplantation in multiple myeloma patients. This study compared anti-PD1 blockage alone to anti-PD1 blockage with a dendritic cell vaccine. Unfortunately, this trial does not have reported results.

Dendritic cells have also been transfected with mature human telomerase reverse transcriptase messenger RNA with the aim of effectively introducing prostate cancer telomerase to the immune system to strongly stimulate a T-cell response (NCT01153113). In a similar fashion, in the clinical trial NCT00514189, the dendritic cell vaccine is loaded with mRNA plus lysate in the treatment of acute myelogenous leukemia. This vaccine design is used to improve the dendritic vaccine´s immunogenicity but makes the vaccine development even more time-consuming (NCT00514189).

### 4.4. DNA Vaccines

DNA neoantigen vaccines share the third-place position with mRNA vaccines as the most common neoantigen vaccine type in ongoing clinical trials. DNA vaccines are rarely combined with an adjuvant due to their high immunogenicity. However, DNA vaccines are relatively commonly combined with checkpoint inhibitors durvalumab or nivolumab. In the clinical trials NCT03199040 and NCT04397003, a common DNA vaccine setting is used by randomizing patients who would receive neoantigen vaccination alone or in combination with durvalumab. GNOS-PV02 is a common DNA vaccine type used in clinical trials, and, for instance, in NCT04251117, it is combined with plasmid-encoded IL-12 and pembrolizumab. The clinical trial NCT03122106, on the other hand, trials DNA neoantigen vaccine as monotherapy following surgical resection and chemotherapy. In this study, a personalized DNA vaccine is designed by integrating prioritized neoantigens and mesothelin epitopes into the pING parent vector. DNA vaccines are also used as a monotherapy in the clinical trial NCT03988283 in the treatment of pediatric patients with brain tumors that are resistant to treatment or have relapsed. In these trials, the intracellular delivery is enhanced with the TDS-IM device, which is a relatively common maneuver seen in multiple DNA vaccine clinical trials.

### 4.5. mRNA Vaccines

Messenger RNA vaccines are a growing category of vaccines. These vaccines have the advantage of not integrating into the cell’s genome [4]. In a similar fashion to DNA vaccines, mRNA vaccines do not use an adjuvant, but instead are often combined with checkpoint inhibitors. Messenger RNA vaccination, called RO7198457, is combined with atezolizumab in the clinical trials NCT04161755 and NCT03289962. However, in NCT03908671 and NCT03468244 mRNA, neoantigen vaccination is used as a monotherapy for advanced esophageal squamous carcinoma, gastric adenocarcinoma, pancreatic adenocarcinoma, and colorectal adenocarcinoma. A study trialing mRNA vaccines in the treatment of gastrointestinal cancer concluded that the vaccine was safe and could potentially be used in combination with checkpoint inhibitors of adoptive T-cell therapy. They detected a mutation-specific T-cell response against the predicted neoantigens and were able to find T-cell receptors that target KRAS mutation. In the same study, nausea and vomiting were reported as side effects with mRNA neoantigen vaccination (NCT03480152).

### 4.6. Other Vaccine Types

There are other vaccine types with a smaller number of clinical trials, such as shared neoantigen vaccines, viral vector vaccines, and fusion protein vaccines. These vaccine types are often new and have the potential to grow in popularity in the future. The clinical trial NCT04041310 trials a viral vector vaccine in the form of a Nous-209 genetic polyvalent vaccine in combination with pembrolizumab. The Nous-209 vaccine in this trial is based on a heterologous prime regimen with great ape adenovirus GAd20-209-FSP for priming and modified vaccinia virus Ankara MVA-209-FSP for boosting. The trial NCT04990479 used a similar vaccine design but combined it with anti-PD-1 immunotherapy in the treatment of stage III/IV melanoma and stage IV lung cancer. Shared neoantigen vaccines are another small vaccine group, and they rely on the fact that some tumor-specific neoantigens are known to be common for a subset of patients. They have the advantage of being faster and cheaper to make than totally personalized neoantigen vaccines but often have fewer specific effects. In the clinical trial NCT03953235, the immunogenicity of shared neoantigen vaccines GRT-C903 and GRT-R904 in combination with nivolumab and ipilimumab is being tested in the treatment of metastatic or advanced lung cancer, colorectal cancer, and pancreatic cancer. Another interesting study is the study NCT03552718, which aims to test a personalized neoepitope yeast-based vaccine, YE-NEO-001, for potentially curatively treated solid cancer. There are unfortunately no results yet from these clinical trials with small vaccine groups.

## 5. Cancer Types

The highest number of clinical trials were aiming to treat lung cancer patients with small-cell lung cancer or non-small-cell lung cancer (Figure 1D). Other common cancer types were melanoma and glioma. The known high neoantigen load is common for these cancer types, which makes them attractive for neoantigen vaccine therapy [3]. These cancer types are the most diagnosed cancers in the world and often lead to death. The poor prognosis is largely due to late diagnosis when the mutation burden is high [16]. For instance, the clinical trial NCT05242965 tested CD105/Yb-1/SOX2/CDH3/MDM2-polyepitope plasmid DNA vaccine (STEMVAC) for stage IV non-squamous non-small-cell lung cancer with the aim of shrinking the tumor and helping the patients with advanced lung cancer (NCT05242965). There are not yet any clinical trials reporting results for lung cancer in our data.

Malign melanoma was the third most researched cancer type in the field of neoantigen cancer vaccines. It evolves from malignant transformation of melanocytes and has a high degree of malignancy and rapid metastasis, leading to poor prognosis [10]. Melanoma has been studied, for instance, in the trial NCT03480152, which illustrated mRNA vaccines as a relative safe alternative treatment that induced a profound T-cell response but did not totally stop disease progression (NCT03480152).

It is not surprising to see gliomas in a high number of the trials, as the life expectancy is generally low and tumors are not often available for surgery. Glioblastoma, the most aggressive form of glioma, also shows a low immune tumor microenvironment, which neoantigen vaccines can attack [10]. Glioma is the cancer type with the most reported results in the field of neoantigen vaccines (NCT0090506) (NCT00293423) (NCT01814813) (NCT02332889) (NCT01280552). For instance, in the trial NCT01280552, neoantigen dendritic cells provided better treatment effects in the treatment of glioblastoma than the control without pulsed neoantigens. The relationship between neoantigen vaccine types and the specific cancer type is difficult to interpret. All vaccine types seemed to be used variably in the treatment of multiple cancer types, and no clear pattern is seen.

## 6. Adjuvants

Adjuvants are generally defined as compounds co-administered with vaccines to improve vaccine efficacy and enhance adaptive immune system induction [17,18]. Adjuvants can strengthen the magnitude, durability, or breadth of the immune response generated by vaccines [19]. A total of 51% of the clinical trials combined neoantigen vaccines with an adjuvant, and eight different adjuvants were used (Figure 2A).

### 6.1. Poly-ICLC

Poly-ICLC was the most commonly used adjuvant. It is a synthetic dsRNA mimic that mimics a product in various viral infections and stimulates innate immunity by promoting pattern recognition receptors, TLR3 and MDA5. These receptors stimulate the activation of cytokines IFN-I and IL-15, enhancing T-cell responses and promoting T-cell expansion [20]. Poly-ICLC is used with a diverse number of neoantigen vaccines. Most commonly, it is used as an adjuvant with peptide vaccines and also mRNA vaccines. Clinical trials NCT03068832, NCT02287428, and NCT03422094 are examples of typical trials using poly-ICLC. In these trials, peptide vaccines are used and administrated in cycles. The first cycle is administrated on days 1, 8, 15, and 22 as the priming phase (except in NCT02287428, where vaccine is also administrated on day 4) following boost cycles. NCT03422094 and NCT02287428 use the NeoVax vaccine, whereas NCT03068832 has another type of peptide vaccine. In NCT03422094, vaccination and poly-ICLC are combined with checkpoint inhibitors nivolumab and ipilimumab, and in NCT02287428, it is combined with pembrolizumab and temozolimde. Another, less typical, trial administrating poly-ICLC is NCT02332889, where poly-ICLC is used with dendritic cell vaccines and even combined with Decitabine adjuvant. In this trial, serious adverse events were observed in the form of disease progression (NCT02332889). Additionally, NCT01635283 combined poly-ICLC with a dendritic cell vaccine in a trial with recurrent low-grade glioma patients. All the patients in this trial stayed alive through the clinical trial but experienced adverse events such as headache, fatigue, and seizures (NCT01635283). FDA has not yet approved poly-ICLC in cancer treatments, and it is still an experimental drug in many cancer types.

### 6.2. Other Adjuvants

GM-CSF (granulocyte macrophage colony-stimulating factor) was the second most common adjuvant used in the clinical trials. This proinflammatory factor is commonly used with peptide and DNA vaccines. By upregulating costimulatory CD80/CD86 molecules and MHC class II, GM-CSF enhances T-cell activation and the function of dendritic cells. Nevertheless, GM-CSF has demonstrated variable success in clinical trials. In some conditions, it has even induced the production of immunosuppressive cells such as regulatory T cells and myeloid-derived suppressor cells, whilst in animal models, GM-CSF has generally shown positive effects [8,21]. GM-CSF was commonly used with a vaccine called iNeo-Vac-P01. In NCT04810910 and NCT03645148, a total of seven doses of GM-CSF are administrated with the vaccine.

QS-21 Stimulon, which was used in several clinical trials, is a saponin-based adjuvant that is known, for instance, from Mosquirix™ malaria vaccine. QS-21 Stimulon leads to the release of Th1 cytokines by acting on antigen-presenting cells. However, QS-21 Stimulon’s mechanism of action is poorly known [22]. QS-21 Stimulon is used in NCT03673020, where it is combined with ASV^®^ AGEN2017 vaccination and in NCT02992977, where it is combined with AutoSynVax. In both of these trials, QS-21 Stimulon is received every other week in subcutaneous injection with the vaccine. These trials have no reported results yet.

Montanide is a water-in-oil emulsion enhancing cytotoxic T-cell responses. It is commonly used in vaccine trials requiring cellular immune response and thus works well with neoantigen cancer vaccines. However, in some clinical trials it has elicited important adverse events. Montanide adjuvant can be used in different forms, such as Montanide ISA-720 and Montanide ISA-51. Only the Montanide ISA-51 subtype was specified in neoantigen vaccine clinical trials [23].

## 7. Combination Therapies

As immune escape remains a problem for neoantigen vaccines´ clinical efficacy, there is a need for combination treatments to overcome the problem. A total of 57% of the clinical trials used combination therapy to boost the neoantigen vaccine’s effectiveness (Figure 2B). The most prevalent group of immunomodulatory antibodies in neoantigen vaccine trials were checkpoint inhibitors. Nivolumab, pembrolizumab, durvalumab, and atezolizumab are amongst the most popular treatments combined with neoantigen vaccines, and they all act by inhibiting PD-1–PD-L1 interaction, which would normally inhibit the immune system’s antitumor reaction [24,25,26,27,28]. Ipilimumab was the third most popular combination treatment in the clinical trials. It is a checkpoint inhibitor that acts by inhibiting CTLA-4, which normally mitigates T-cell activity [28]. Temozolomide, on the other hand, is largely utilized in malignant brain tumors and is a common first-line treatment in glioblastoma [29,30]. NCT00905060 tested a heat shock protein vaccine in combination with temozolomide for patients with brain and central nervous system tumors. In this trial, the median overall survival was 23.8 months, whereas the median progression-free survival was 18 months. Combining neoantigen vaccine with standard therapy, such as temozolomide, was also demonstrated to be crucial, and it had the potential to improve survival for glioblastoma patients (NCT00905060).

### Chemotherapy, Radiation Therapy, and Surgery

Chemotherapy, radiation therapy, and surgery are mainstays for cancer treatment, and that is also the case with neoantigen vaccine trials. All these three forms are commonly used neoadjuvants in neoantigen vaccine treatment. In multiple clinical trials, one of the inclusion criteria was that the patient did not answer well to earlier chemotherapy. Chemotherapy and radiation therapy were also in some cases used as adjuvant treatments for neoantigen vaccines. The decreased expression of neoantigens in tumors is a challenge with neoantigen vaccines. This can be addressed by vaccines that contain multiple neoepitopes and combining them with DNA-damaging chemotherapy. Chemotherapy can introduce new somatic mutations in tumors and add to the efficacy of neoantigen vaccines. Thus, chemotherapy can be essential with neoantigen vaccines [30]. Surgery, on the other hand, is often required for neoantigen vaccine creation and therefore obligatory in many cases for neoantigen vaccine treatment.

## 8. Vaccinations

NeoVax, as the most commonly used vaccine, was used in one-tenth of the clinical trials. It is an investigational peptide vaccine that is frequently boosted with poly-ICLC and is based on research at the Dana Farber Cancer Institute. NeoVax has been trialed for various different cancer types, glioblastoma patients being the most typical target group. For instance, in NCT02287428, NeoVax is trialed in combination with radiation therapy, pembrolizumab, and temozolomide for newly diagnosed glioblastoma patients, whereas in NCT01970358, NeoVax is used as a monotherapy for treating melanoma patients. In the existing results, NeoVax has been shown to be a safe and feasible treatment that, in combination with anti-PD-1, is at its best even able to induce complete tumor regression [31].

Another vaccine type that has been able to show promising results is the HSPPC-96 and -97 neoantigen vaccines. HSPPC-96 and -97 are two heat shock protein vaccines that, in a similar fashion to NeoVax, are mostly trialed in the treatment of glioblastoma. In the clinical trialNCT03018288, HSPPC-96 is combined with temozolomide and pembrolizumab in the treatment of glioblastoma after surgery, while in another trial, NCT00293423, HSPPC-96 was trialed as monotherapy after surgical resection. This trial showed that 66% of the patients developed an immunological response after HSPPC-96 vaccination (NCT00293423). Median overall survival in this study was 42.6 weeks, but unfortunately, there was no control group in the study, so the results are difficult to compare with those of other treatments (NCT00293423). HSPPC-96/-97 neoantigen vaccination research has shown the vaccine to be a safe treatment for various cancer types, but the efficiency of the vaccination was partially disappointing.

Autogene cevumeran, also known as RO7198457, is another vaccine type that is worth mentioning here. It is an mRNA-based cancer vaccine that was developed by Genentech. It is less commonly trialed for glioblastoma patients than the other mentioned vaccination types, and it seems to have a more diverse cancer target range, varying from bladder cancer to non-small lung cancer. Autogene cevumeran has not yet shown results, but it is often tested in combination with checkpoint inhibitors such as atezolizumab or pembrolizumab (NCT03289962) (NCT03815058) (NCT04267237) (NCT03289962).

Neoantigen vaccines can be administrated in various ways: subcutaneously, intramuscularly, intraperitoneally, intradermally, intravenously, or intranodally. Subcutaneous and intramuscular injections were the most common administration methods. Intramuscular administration was often combined with an integrated electroporation device, such as the TDS-IM system, which can improve the delivery. Subcutaneous administrations were mainly used for peptide vaccines while DNA vaccines were often administrated intramuscularly. Dendritic cell vaccines, on the other hand, showed great variation in the administration methods, although subcutaneous injection was the most recurring approach. In the clinical trial NCT04397003, for instance, two DNA vaccine injections were performed as intramuscular injections with the TDS-IM system into the deltoideus or lateralis (NCT04397003). In the clinical trial NCT04266730, on the other hand, peptide vaccine PANDA-VAC was administrated subcutaneously with poly-ICLC via three equal-volume injections in an atm and one in each leg.

## 9. Discussion

In total, 147 neoantigen vaccine clinical trials were analyzed. In as many as 41% of the clinical trials, peptide vaccines were used, whereas 17% of the clinical trials used dendritic cell vaccines, 10% DNA vaccines, and 10% used mRNA vaccines. A total of 51% of the clinical trials used an adjuvant, with poly-ICLC being the most common adjuvant. In these clinical trials, the patient most commonly suffered from glioblastoma, small-cell lung cancer, non-small lung cancer, and malignant melanoma, comprising more than half of the clinical trials. Importantly, neoantigen vaccines were effectively combined with other cancer treatments, and the checkpoint inhibitors nivolumab, pembrolizumab, and ipilumamb were the most recurring combinatory treatments.

### 9.1. Vaccine Types

Peptide vaccines were the clear dominating group of neoantigen vaccines. While short-peptide vaccines were more common in older clinical trials, the long-peptide vaccines represented a larger group in later clinical trials. Intriguingly, this illustrates a common trend in neoantigen vaccine development in which the vaccines become more complex over time as the simpler vaccine forms show inadequate effects. The reason behind the prevalence of peptide vaccines could be that they are well-tolerated by patients, simple to manufacture, and stable in storage [8,32]. Furthermore, peptide neoantigen vaccines are relatively cost-effective and thus likely to remain the most favorable choice of most studies in the future [8]. Additionally, peptide vaccines have been able to generate tumor-infiltrating T-cell responses even in immunologically cold tumors [33]. However, mRNA vaccines are a growing vaccine group that at the moment is seen in 10% of clinical trials. Messenger RNA vaccines have the advantage of not integrating into the host genome, as RNAs are translated in cytosol so it does not have to penetrate the nuclear cell membrane, and therefore have a high safety profile [34]. Nevertheless, it is also questionable if DNA vaccines integrate into the genome in DNA vaccination. There is an increasing amount of evidence confirming that DNA vaccines have an extremely low probability of integrating into the human genome [35]. Additionally, RNA vaccine generation is fast and effective compared to that of peptide vaccines. On the other hand, mRNAs are easily degraded by extracellular RNases, which are tackled by using protective strategies, such as liposome and nanoparticle encasement [3]. RNA vaccines’ downsides are the storage and lifetime challenges, and additionally, the risk for potentially harmful type I interferon (IFN) responses [34]. Nevertheless, given the benefits of RNA vaccines, this neoantigen vaccine type is becoming increasingly popular. Interestingly, the vaccine type seen in the highest number of clinical trials is peptide vaccines instead of RNA vaccines, probably owing to the peptide vaccines´ low production costs and better durability against degradation.

### 9.2. Cancer Types

Various brain cancers were the most researched cancer types in neoantigen vaccine clinical trials, comprising 22% of all cancer types. A total of 91% of the brain cancers were different gliomas, half of which were glioblastomas. The remaining 9% of the brain cancers represented pediatric brain cancers. Intriguingly, the single most dominating cancer groups were small-cell lung cancer, constituting 13% of the cancer types, and non-small lung cancer with another 13%. In total, 12% of malignant melanoma was additionally another big cancer group. Common for all these cancer groups is their high mortality and the acute need for new effective treatments. The cancers located in the brain are especially interesting cancer types, as neoantigen vaccines need to activate T cells that succeed in migrating over the blood–brain barrier [7].

Peptide vaccines represented the largest vaccine type in all these big cancer types. Surprisingly, there was no clear correlation between vaccine types and cancer types, and all the vaccine types were being tested for the majority of the big cancer types. From the clinical trials reporting results, the overall trend seems to be that neoantigen vaccination is more effective for early-stage tumors while effects are often weak for patients with late-stage tumors [9]. This is unfortunate as neoantigen vaccines were originally often thought to provide help for relatively better treatment of advanced cancer types. The results from the clinical trials show that disease-free survival time can in many cases be increased by neoantigen vaccines even in late-stage tumors, but neoantigen vaccines impart the most benefit if used earlier. Unfortunately, neoantigen vaccines are often used when patients have tumor relapse or do not respond to treatments. This is because neoantigen vaccines are expensive and time-consuming to create and are, therefore, often used as a last hope rather than an early treatment option. However, if neoantigen vaccines had been administered earlier, these patients might not have relapsed or would have shown better treatment responses. Therefore, it is reasonable to administer neoantigen vaccines earlier in the treatment to prevent relapse, as seen in some of the clinical trials. The problem is knowing which patients will benefit most from the neoantigen vaccine treatment as the expensive and resource-consuming treatment can often be provided only for a limited number of patient groups. It is probable that neoantigen vaccines will be more extensively used for early-stage tumors in future clinical trials.

### 9.3. Combination Therapies

The current challenge with neoantigen vaccines is their long production time, and especially during that time, other treatments are often necessary. Furthermore, neoantigen vaccines cannot likely eliminate malignant tumors alone due to tumors’ rapidly changing neoantigen expression. Therefore, it is not surprising that in 58% of the clinical trials, neoantigen vaccines were combined with another medical treatment (not including chemotherapy, radiation therapy, or surgery). Most commonly, these treatments were checkpoint-inhibiting antibodies such as PD-L1 or CTLA-4 inhibitors repressing the tumor´s immune escape mechanisms. The most recurring combinatory treatments were nivolumab, seen in 15% of the clinical trials, pembrolizumab in 12%, ipilimumab in 11%, and temozolomide in 5%. Neoantigen vaccines induce specific T-cell production, which upregulates PD-L1 expression. Checkpoint inhibitors solve this problem by removing the T-cell inhibition and letting vaccination work optimally [9]. According to the available results from the clinical trials, combining neoantigens with checkpoint inhibitors improves the patients´ overall survival [10].

Furthermore, the traditional treatments of chemotherapy and radiotherapy have the ability to increase the number of antigens released by cancer, which enhances vaccination effects. Most of the patients in the clinical trials had received chemotherapy or radiotherapy prior to neoantigen vaccine treatment. Additionally, surgery is often required in the creation of the neoantigen vaccines, and therefore, surgery is often also part of the standard-of-care treatments prior neoantigen vaccine administration.

### 9.4. Adjuvants

In the clinical trials, as many as 51% of the trials used an adjuvant to boost the vaccination effects. Strikingly, poly-ICLC was clearly the most popular adjuvant of choice, being in 30% of the clinical trials. This is not surprising, as poly-ICLC is commonly used as an immune stimulator with peptide vaccines, which were the most popular vaccine type in the clinical trials. However, poly-ICLC was also used with mRNA, dendritic cells, and DNA vaccines, which illustrates poly-ICLC´s wide usage spectrum. GM-CSF was the second most popular adjuvant choice as it was used in 7% of the trials. It has a much more limited diversity when it comes to different vaccine types, and it is used with peptide vaccines or with DNA vaccines. Generally, mRNA vaccines are expected to be the group of vaccines requiring the least usage of an adjuvant. However, in the clinical trials, dendritic cell vaccines were the group that proportionally required the least usage of an adjuvant. From the currently limited data, it is difficult to estimate if the usage of an adjuvant is beneficial in neoantigen vaccination. Half of the reported studies have not used an adjuvant, but the current trend is that an adjuvant is more commonly used in later studies.

### 9.5. Limitations

The major limitation in neoantigen vaccine treatment is still the expensive and time-consuming development process that is required to create personalized neoantigens. Combined antigens that are common for certain patient subgroups have been used to overcome this problem but with nonoptimal results (NCT03953235). Peptide vaccines are often the cheapest neoantigen vaccine type to develop while mRNA vaccines are more time-efficient [3]. As neoantigen vaccines are often used in the treatment of late-stage cancer, patients do not necessarily have time to wait for neoantigen production. A solution to this problem would be to start with neoantigen vaccine therapy earlier in the disease progression, which, however, would require a high number of patients to be included in the neoantigen vaccine therapies and therefore high expenses. Another limitation in neoantigen vaccine treatment is the variable treatment outcomes, which make it difficult to predict when neoantigen vaccines should be administered.

## 10. Conclusions

Neoantigen vaccination is a rapidly developing cancer treatment with an increasing number and diversity of clinical trials. One of the most significant challenges with neoantigen vaccines is tumors’ changing antigen expression. The different combination treatments can act as a solution to this problem, and it is highly important that breakthroughs are seen in combining neoantigens with existing treatments. Neoantigen vaccines are most commonly peptide vaccines combined with a poly-ICLC adjuvant, but in recent years, neoantigen vaccines have grown in diversity. The most researched cancer types are lung cancers, gliomas, and melanoma, and these cancer types constitute more than half of the current clinical trials. Neoantigen vaccination seems to work best together with standard therapies for early-stage tumors, but provides less optimal results with more advanced tumors. In the future, neoantigen vaccines are likely to develop further and become substantially helpful as a cancer treatment, but further understanding of tumors´ immunosuppressive milieu and immune escape mechanisms is crucial. It is important to note that the majority of the current clinical trials are active and waiting for the results, and it is, therefore, likely that there are a growing number of data that will be provided in the near future.

## Figures and Tables

**Figure 1 cancers-14-05163-f001:**
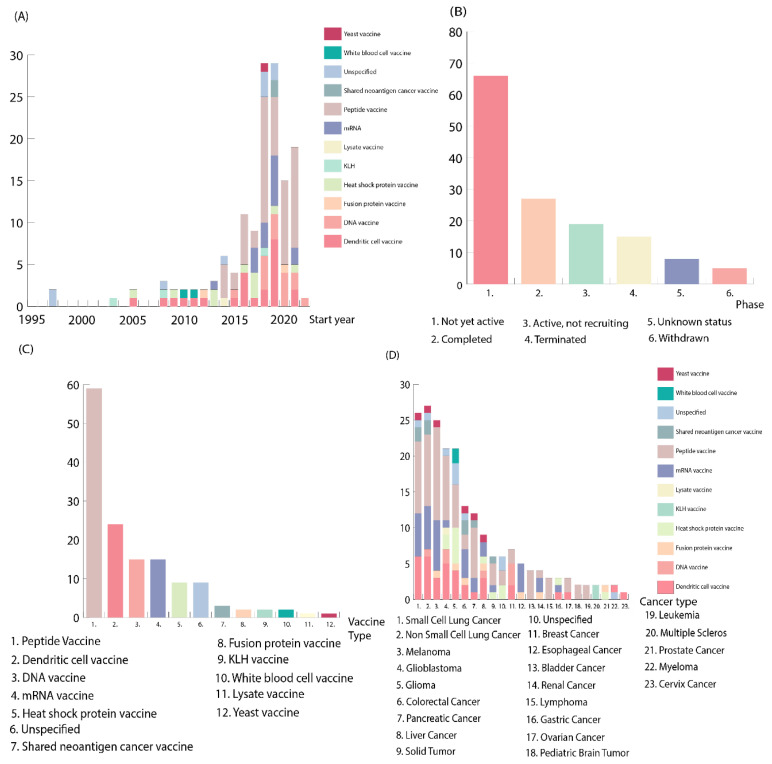
(**A**) Number of neoantigen vaccine clinical trials registered in ClinicalTrials.gov per year. Colors correspond to different neoantigen vaccine types, whereas each bar corresponds to the number of new clinical trials registered each year. The number of clinical trials increased consistently, reaching a peak in 2019. The year 2022 has a small number of clinical trials as it has not been completed at the time of writing. (**B**) The current phase of the neoantigen vaccine clinical trials. *Y* axis corresponds to the number of clinical trials in each phase, whereas the different phases are labeled on the *X* axis. The first bar, the Not yet active group, includes clinical trials that are recruiting, not yet recruiting, and enrolling by invitation, which can partially explain the peak. Suspended clinical trials are included in the Withdrawn group. (**C**) The number of neoantigen vaccine types in the clinical trials. Each bar corresponds to the number of clinical trials registered, while the number on the *X* axis corresponds to the different neoantigen vaccine types. Vaccine types were selected according to what is listed in the registration in Clinical.Trials.gov. Neoantigen vaccines are typically divided into just 4 categories, but in our analysis the additional groups are also provided to give a better picture of the diversity of the neoantigen vaccine clinical trials. (**D**) The number of cancer types in the registered clinical trials corresponding to neoantigen vaccine type used in the treatment. Colors correspond to different vaccine types, and the numbers on *X* axis correspond to the different cancer types. Cancer types are divided according to the place of origin. The largest cancer groups are further divided into smaller groups. Lung cancer is divided into small-cell lung cancer and non-small-cell lung cancer, and glioblastoma is included as a separate group, instead of including it in the glioma group. Multiple clinical trials included multiple cancer types, and each clinical trial is therefore included in several cancer type categories.

**Figure 2 cancers-14-05163-f002:**
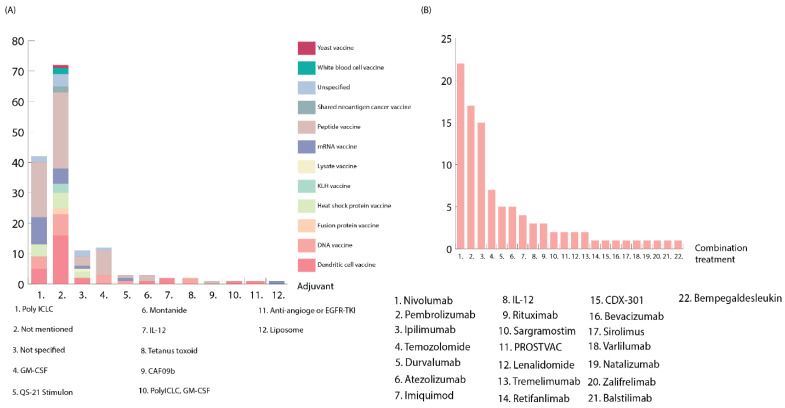
(**A**) Adjuvants used in combination with different neoantigen vaccine types. Each bar illustrates the number of clinical trials using the specific adjuvant, while the *X* axis illustrates the different adjuvants. Colors illustrate the different vaccine types. (**B**) Combinatorial treatments with neoantigen vaccines. Each bar corresponds to the number of clinical trials with combinatorial treatments and the *X* axis corresponds to the different combinatorial treatments. Combinatorial treatments do not include chemotherapy, radiation therapy, or surgical operations. Checkpoint inhibitors, such as PD-L1 and CTLA-4 inhibitors, are the most popular combinatorial treatments using neoantigen vaccines.

## Data Availability

All data tables are available from the authors upon request.

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
