# Peer review of "Neoantigen Vaccines; Clinical Trials, Classes, Indications, Adjuvants and Combinatorial Treatments"

_cancers, 2022, doi:10.3390/cancers14205163_

Round 1

Reviewer 1 Report

The manuscript by Schiöth and coworkers is an interesting and well-written summary on neoantigens vaccination, with a particular focus on clinical trials. I have some issues:

- I think that an explanatory table with all/some of the clinical trials discussed in this review will enhance the manuscript and facilitate its reading.

-The introduction and, in general, the review is quite similar to this recent one: https://www.nature.com/articles/s41571-020-00460-2

It might be of interest if the authors remark the novelties of their manuscript.

-The authors would like to comment more deeply on the limitations of this personalized treatments (high costs and time).

-The information shown in Figure 1C and 1D is not commented in the Results section, which is fine, because it is discussed in other sections. Maybe the authors could divide this figure into Vaccine types and Cancer types, or they could mention these figures whenever it is needed.

- Peptide vaccines: The authors might include and/or discuss: clinical trial NCT03012100: Multi-epitope Folate Receptor Alpha Peptide Vaccine, GM-CSF, and Cyclophosphamide in Treating Patients With Triple Negative Breast Cancer or NCT02600949: Personalized Peptide Vaccine in Treating Patients With Advanced Pancreatic Cancer or Colorectal Cancer

- Heat shock protein vaccines section, the don’t mention a new clinical trial of

GP96 Heat Shock Protein-Peptide Complex Vaccine in Treating Patients With Liver

Cancer (NCT04206254 Not yet recruiting).

- The dendritic vaccine section is rather short. There are other clinical trials in which they block anti-PD1 together with the dendritic cell / myeloma vaccines following stem cells transplantation (NCT01067287). Moreover, there was a Phase II Study of Active Immunotherapy with Mature, Human Telomerase Reverse Transcriptase Messenger RNA -Transfected, Autologous Dendritic Cells (DC) Administered In A Prime-Boost Format to Subjects With Metastatic Prostate Cancer (NCT01153113 Withdrawn).

-There are at least two different strategies involving DCs that can be used to promote antitumor immunity: in situ vaccination and canonical vaccination and this is not discussed here. Differences in DC subsets might also be discussed: Monocyte-derived DCs (mo-DCs) and leukemia-derived DCs (DCleu) are the main types of DCs used in vaccines for AML and MDS thus far.

-line 265: “The second and third most popular neoantigen vaccine candidates are dendritic cell-based vaccines”. Second and third? Is this a mistake?

-DNA vaccine clinical trial: Neoepitope-based Personalized DNA Vaccine Approach in Pediatric Patients With Recurrent Brain Tumors (NCT03988283) is mentioned, but not discussed.

-There is another clinical trial that combine the mRNA plus lysate loaded dendric cell vaccines to acute myelogenous leukemia (AML) (NCT00514189) that can be included.

- line 209. there is a clinical trial that use A Multiple Antigen Vaccine

(STEMVAC) for the Treatment of Patients With Stage IV Non-Squamous Non-Small Cell Lung Cancer (NCT05242965).

-Currently, there are more adjuvants for cancer vaccines that the ones mentioned in this manuscropt, see this review on this issue: https://doi.org/10.3389%2Ffimmu.2020.615240

- Differences due to the administration route of the different vaccines are not discussed. Maybe can be mentioned in section 4.12.

-Section 4.12. Lines 328-338 comment on HSPPC-96 vaccine. These sentences repeat partially section 4.3. This repetition should be avoided.

Minor:

-Why is the number of the clinical trial written after the dot of the sentence referring to it, and not before the dot? e.g. line 150: “The median  progression free survival was observed to be 18 months with adverse events such as anemia, fatigue, and hyperglycemia. (NCT00905060)”

- line 171: “Also, the progression free survival was higher for the group receiving dendritic cells pulsed with antigens indicating it as a potentially better treatment alternative.”  I think that this sentence is confusing and might be rephrased.

-Line 368: Is there any actual evidence of DNA integration upon DNA vaccination?

-Line 376: “Interestingly, the most researched vaccination type that has provided the most promising data, is not a mRNA vaccine, but a peptide vaccine.” I think this sentence requires a reference.

Author Response

We thank the reviewer for the thorough comments on our manuscript; below you can find all the answers to your comments:

-I think that an explanatory table with all/some of the clinical trials discussed in this review will enhance the manuscript and facilitate its reading.

We have provided a table with all the clinical trials. We suggest it be included in the supplementary material. We selected to show in the table only the columns that are relevant to the results to avoid the table to become too big. A more detailed table will be available from the authors if needed. The table is large as all the analyzed clinical trials are included. Therefore, we provide the table as an Excel file.

-The introduction and, in general, the review is quite similar to this recent one: https://www.nature.com/articles/s41571-020-00460-2

It might be of interest if the authors remark on the novelties of their manuscript.

The following remarks about the novelties of the manuscript are added in the introduction:

“To our best knowledge, there are no published review articles about neoantigen vaccine clinical trials that would extensively cover the whole field. Therefore, we choose to include all the available neoantigen vaccine clinical trials and quantitatively evaluate their features, such as status, results, and indications. Here we provide a detailed analysis of neoantigen vaccines in cancer treatment from the first trial in 1997 until today. We include neoantigen vaccine types that are commonly not included as their own group and show how trends in the popularity of different vaccine types develop over time. We also show how cancer types vary for different indications and resonate with potential reasons for the popularity of certain vaccines. Furthermore, we illustrate the usage of adjuvants therapies and combinatorial treatments.”

Compared to the review published in Nature, our review provides more quantitative information about the trends in neoantigen vaccine development by systematically comparing 142 different neoantigen vaccine clinical trials and by providing detailed illustrations of their properties. The review article in nature visualizes how the neoantigen vaccines are developed while our review concentrates more on how neoantigen vaccines are used in practice.

-The authors would like to comment more deeply on the limitations of these personalized treatments (high costs and time).

A limitation section is added at the end of the discussion to highlight the preserving limitations of neoantigen vaccines:

“The major limitation in the neoantigen vaccine treatment is still the expensive and time-consuming development process which is required to create personalized neoantigens. Share antigens that are common for certain patient subgroups have been used to overcome this problem but with low results (NCT03953235). Peptide vaccines are often the cheapest neoantigen vaccine type to develop while mRNA vaccines are more time efficient [3, 25]. As neoantigen vaccines are often used in the treatment of late-stage cancer, patients do not necessarily have time to wait for neoantigen production. A solution for this problem would be to start with neoantigen vaccine therapy earlier in the disease progression which however would require a high number of patients to be included in the neoantigen vaccine therapies and therefore high expenses. Another limitation in the neoantigen vaccine treatment is the variable treatment outcomes that make it difficult to predict when neoantigen vaccines are reasonable to set in.”

-The information shown in Figure 1C and 1D is not commented in the Results section, which is fine, because it is discussed in other sections. Maybe the authors could divide this figure into Vaccine types and Cancer types, or they could mention these figures whenever it is needed.

Brackets with relevant figure labels are added in each section to facilitate easier reading for the manuscript's readers to interpret when the information from the figures is discussed, such as: “The clear majority of the clinical trials use peptide neoantigen vaccines (Figure 1C).”

- Peptide vaccines: The authors might include and/or discuss: clinical trial NCT03012100: Multi-epitope Folate Receptor Alpha Peptide Vaccine, GM-CSF, and Cyclophosphamide in Treating Patients With Triple Negative Breast Cancer or NCT02600949: Personalized Peptide Vaccine in Treating Patients With Advanced Pancreatic Cancer or Colorectal Cancer

The clinical trials NCT03012100 and NCT02600949 are added in the peptide vaccine section:

“Other interesting peptide vaccine clinical trials include NCT03012100 which tests multi-epitope folate receptor alpha peptide vaccine combined with sargramostim (GM-CSF) and cyclophosphamide chemotherapy in treating triple-negative breast cancer. These vaccines are created from a person´s white blood cells that are mixed with tumor proteins. In this case, the vaccination is aimed at preventing disease relapse after surgery and other standard treatments. (NCT03012100) This illustrates the variability in peptide vaccine treatment aims, as many neoantigen vaccine clinical trials use peptide vaccines to directly attack uncontrollably growing tumors instead of preventing relapse. In the clinical trial NCT02600949, for instance, peptide vaccines are trialed for the treatment of spread pancreatic or colorectal cancers that cannot normally be cured or controlled with available treatments. In this clinical trial peptide vaccine is combined with imiquimod, pembrolizumab, and sotigalimab. (NCT02600949)”

Additionally, these trials are also added to the results.

- Heat shock protein vaccines section, the don’t mention a new clinical trial of

GP96 Heat Shock Protein-Peptide Complex Vaccine in Treating Patients With Liver

Cancer (NCT04206254 Not yet recruiting).

NCT04206254 is added in the heat shock vaccine section:

“Other heat shock protein vaccine types include gp96 which is trialed as a post-operation treatment for liver cancer patients in NCT04206254 or rHSC-DIPGVax which is mainly used for childhood diffuse intrinsic pontine gliomas or diffuse midline gliomas (NCT04206254) (NCT04943848).”

This trial is also added to the results.

- The dendritic vaccine section is rather short. There are other clinical trials in which they block anti-PD1 together with the dendritic cell/myeloma vaccines following stem cells transplantation (NCT01067287). Moreover, there was a Phase II Study of Active Immunotherapy with Mature, Human Telomerase Reverse Transcriptase Messenger RNA -Transfected, Autologous Dendritic Cells (DC) Administered In A Prime-Boost Format to Subjects With Metastatic Prostate Cancer (NCT01153113 Withdrawn).

The clinical trials NCT01067287 and NCT01153113 are added in the dendritic cell vaccines section:

“The study NCT01067287, on the other hand, trialed dendritic cell vaccines with anti-PD1 blockage following stem cell transplantation in multiple myeloma patients. This study compared anti-PD1 blockage alone to anti-PD1 blockage with a dendritic cell vaccine. Unfortunately, this trial does not have reported results.”

“Dendritic cells have also been transfected with mature human telomerase reverse transcriptase messenger RNA to be able to effectively present prostate cancer telomerase to the immune system to strongly stimulate T cell response (NCT01153113).”

These trials are added to the results.

-There are at least two different strategies involving DCs that can be used to promote antitumor immunity: in situ vaccination and canonical vaccination and this is not discussed here. Differences in DC subsets might also be discussed: Monocyte-derived DCs (mo-DCs) and leukemia-derived DCs (DCleu) are the main types of DCs used in vaccines for AML and MDS thus far.

The dendritic cell vaccine section is elaborated by adding the following section:

“There are multiple dendritic cell subsets available for vaccination where monocyte-derived dendritic cells and leukemia-derived dendritic cells are the two main types. Dendritic cell vaccines can be used to promote antitumor immunity by using different strategies, such as in situ vaccination and in vitro canonical vaccination. In situ vaccination promotes dendritic cell uptake and tumor recognition by releasing tumor antigens locally through standard therapies whereas, canonical vaccination loads tumor anti-gens into dendritic cells before delivering dendritic cells to patients [18].”

-line 265: “The second and third most popular neoantigen vaccine candidates are dendritic cell-based vaccines”. Second and third? Is this a mistake?

Yes. The line is now changed to: “The second most popular neoantigen vaccine candidates are dendritic cell-based vaccines.”

-DNA vaccine clinical trial: Neoepitope-based Personalized DNA Vaccine Approach in Pediatric Patients With Recurrent Brain Tumors (NCT03988283) is mentioned, but not discussed.

NCT03988283 is added in the DNA vaccine section:

“DNA vaccines are used as a monotherapy also in clinical trial NCT03988283 in the treatment of pediatric patients with brain tumors that are resistant to treatment or have relapsed. In these trials, the intracellular delivery is enhanced with TDS-IM device which is a relatively common maneuver in multiple DNA vaccine clinical trials.”

This trial is also added to the results.

-There is another clinical trial that combine the mRNA plus lysate loaded dendric cell vaccines to acute myelogenous leukemia (AML) (NCT00514189) that can be included.

NCT00514189 is added in the dendritic cell vaccine section:

“In a similar fashion, in the clinical trial NCT00514189, dendritic cell vaccine is loaded with mRNA plus lysate in the treatment of acute myelogenous leukemia. This vaccine design is used to improve dendritic vaccine´s immunogenicity but makes the vaccine development even more time consuming (NCT00514189).”

This trial is also added to the results.

- line 209. there is a clinical trial that use A Multiple Antigen Vaccine

(STEMVAC) for the Treatment of Patients With Stage IV Non-Squamous Non-Small Cell Lung Cancer (NCT05242965).

The following section is added in the cancer type section discussing lung cancer:

“For instance, the clinical trial NCT05242965 tested CD105/Yb-1/SOX2/CDH3/MDM2-polyepitope plasmid DNA vaccine (STEMVAC) for stage IV non-squamous non-small cell lung cancer with the aim to shrink the tumor and help the patients with an advanced lung cancer (NCT05242965).”

This trial is also added to the results.

-Currently, there are more adjuvants for cancer vaccines than the ones mentioned in this manuscropt, see this review on this issue: https://doi.org/10.3389%2Ffimmu.2020.615240

All the clinical trials are reviewed again and IL-12, tetanus toxoid, and liposomal administration are added as adjuvants although these were not accepted as adjuvants in our earlier review. All these adjuvants are added to the diagram and all the results were modified according to these changes.

- Differences due to the administration route of the different vaccines are not discussed. Maybe can be mentioned in section 4.12.

A paragraph about vaccine administration is added in section 4.12.:

“Neoantigen vaccines can be administrated in various ways: subcutaneously, intramuscularly, intraperitoneally, intradermally, intravenously, or intranodally. Subcutaneous and intramuscular injections were the most common administration ways. Intramuscular administration was often combined with an integrated electroporation device, such as TDS-IM system that can improve the delivery. Subcutaneous administrations were mainly used for peptide vaccines while DNA vaccines were often administrated intramuscularly. Dendritic cell vaccines, on the other hand, showed great variation in administration way although subcutaneous injection was the most recurring administration way. In the clinical trial NCT04397003, for instance, two DNA vaccine injections were performed as intramuscular injections with the TDS-IM system into deltoideus or lateralis (NCT04397003). In the clinical trial, NCT04266730, on the other hand, peptide vaccine PANDA-VAC were administrated subcutaneously with Poly-ICLC via 3 equal volume injections in an atm and one in each leg NCT04266730.”

-Section 4.12. Lines 328-338 comment on HSPPC-96 vaccine. These sentences repeat partially section 4.3. This repetition should be avoided.

This repetition is avoided by changing the section in 4.12 from this:

“Another vaccine type that has been able show promising results are HSPPC -96 and -97 neoantigen vaccines. HSPPC -96 and -97 are two heat shock protein vaccines that in a similar fashion as NeoVax, are mostly trialed in the treatment of glioblastoma. In the clinical trial NCT01814813, HSPPC -96 is combined with bevaciumab. Discouragingly the results show that bevaciumab alone has better effects than when combined with heat shock proteins vaccination. Another clinical trial showed that 66% of the patients developed an immunological response after HSPPC -96 vaccination.”

to this:

“Another vaccine type that has been able to show promising results are HSPPC -96 and -97 neoantigen vaccines. HSPPC -96 and -97 are two heat shock protein vaccines that in the similar fashion as NeoVax, are mostly trialed in the treatment of glioblastoma. In the clinical trial NCT03018288, HSPPC -96 is combined with temozolomide and pembrolizumab in the treatment of glioblastoma after surgery while in another trial, NCT00293423, HSPPC-96 was trialed as monotherapy after surgical resection. This trial showed that 66% of the patients developed an immunological response after HSPPC-96 vaccination (NCT00293423).”

Minor:

-Why is the number of the clinical trial written after the dot of the sentence referring to it, and not before the dot? e.g. line 150: “The median progression free survival was observed to be 18 months with adverse events such as anemia, fatigue, and hyperglycemia. (NCT00905060)”

The number of the clinical trial is moved before the dot in all cases.

- line 171: “Also, the progression free survival was higher for the group receiving dendritic cells pulsed with antigens indicating it as a potentially better treatment alternative.” I think that this sentence is confusing and might be rephrased.

The line 171 is changed from this:

“Also, the progression free survival was higher for the group receiving dendritic cells pulsed with antigens indicating it as a potentially better treatment alternative.”

to this:

“In a similar fashion, also the progression free survival was higher for the dendritic cell pulsed with antigen than for the dendritic cells not pulsed with antigens. According to this trial, dendritic cells with pulsed antigens would be a more effective treatment alternative. (NCT01280552).”

-Line 368: Is there any actual evidence of DNA integration upon DNA vaccination?

The following sentences are added after line 368 to make it clear that the risk of integration in the genome even with DNA vaccines is actually low:

“Nevertheless, it is also questionable if DNA vaccines integrate into the genome in DNA vaccination either. There is an increasing amount of evidence confirming that DNA vaccines have an extremely low probability of integrating into the human genome [39].”

-Line 376: “Interestingly, the most researched vaccination type that has provided the most promising data, is not a mRNA vaccine, but a peptide vaccine.” I think this sentence requires a reference.

This conclusion is actually drawn by us based on the reported results from the clinical trials. We have changed the sentence to a milder format:

“Interestingly, the vaccine type with the highest number of clinical trials is peptide vaccines instead of RNA vaccines, probably owing to the peptide vaccines' low production costs and better durability against degradation.”

Reviewer 2 Report

The manuscript is well written and should be of great interest to those in the field in general.  

Author Response

We thank the reviewer for the comments on our manuscript.

We have done language modifications throughout the article to improve readability and correct language.

Reviewer 3 Report

In this study, Niemi et al. conducted a search on ClinicalTrials.gov to examine all the Clinical trials on cancer neoantigen vaccines and analyzed them in terms of vaccine type, cancer type, adjuvant and combinatorial treatments. Although the topic is useful to provide an overview on the developments in the field, the way in which the informations are presented is not so well organized, sometimes the concepts are redundant and lacking of constructive discussion on the topics.

Comments

·       A more comprehensive overview on the neoantigen vaccine types should be provided, including some categories of vaccine currently missing (for example viral vectored vaccines, see trial NCT04041310, NCT04990479, NCT04041310, etc).The potential advantages and disadvantages of the different types of neoantigen vaccines should be discussed, in the context of the recent clinical results when available (immunogenicity of the different vaccine platforms, feasibility, proof of principles for T cell activity/tumor infiltration, other).

·       The authors state that from the clinical trial reports, the overall trend seems to be that neoantigen vaccination is more effective for early-stage tumors while effects are often weak for late stage tumors. However, there is no discussion on this important point, neither on the potential implications for the field (future trials, etc).

·       The paragraph 4 “Vaccine types” includes sub-paragraphs not always coherent with the topic of the title. For example, the sub-paragraph 4.6 “Cancer Types” on tumor indications is under the section “Vaccine types”. The same applies to the sub-paragraph 4.8 “Poly ICLC”, which could go under the “Adjuvants” paragraph. Sub-paragraph 4.12 “Vaccination” is redundant, discussing points already presented in previous sections.

·       Please, revise some typos/ grammatically incorrect sentences, such as:

o   Paragraph 4.2: “HSPPC-96 vaccine has for instance in combination with temozolomide used in the treatment of glioblastoma”

o   in the conclusion paragraph: “In future, neoantigen vaccines are likely to develop further and become substantial help as a cancer treatment but further understanding of tumors´ immunosuppressive milieu and immune escape mechanisms are.” Are what? Please, complete the sentence.

·       The graphs in figure 1 and 2 should include y axes label. The use of the same color code for the same vaccine category shown across figures would be preferable (Fig1A and 2A).

Author Response

We thank the reviewer for the thorough comments on our manuscript; below you can find all the answers to your comments:

- A more comprehensive overview on the neoantigen vaccine types should be provided, including some categories of vaccine currently missing (for example viral vectored vaccines, see trial NCT04041310, NCT04990479, NCT04041310, etc).The potential advantages and disadvantages of the different types of neoantigen vaccines should be discussed, in the context of the recent clinical results when available (immunogenicity of the different vaccine platforms, feasibility, proof of principles for T cell activity/tumor infiltration, other).

A new subtitle “Other vaccine types” is added under section 4 to provide a more comprehensive overview of the different neoantigen vaccines. Viral vector vaccines are added in this section and also other small vaccine groups are taken up in this section:

“4.6. Other vaccine types

There are other vaccine types with a smaller number of clinical trials, such as shared neoantigen vaccines, viral vector vaccines, and fusion protein vaccines. These vaccine types are often new and have the potential to grow in the future. The clinical trial, NCT04041310, trials a viral vector vaccine in the form of a Nous-209 genetic polyvalent vaccine in combination with pembrolizumab. The Nous-209 vaccine in this trial is based on a heterologous prime regimen with Great Ape Adenovirus GAd20-209-FSP for priming and Modified Vaccinia virus Ankara MVA-209-FSP for boosting. The trial NCT04990479, used a similar vaccine design but combined with anti-PD-1 immunotherapy in the treatment of stage III/IV melanoma and stage IV lung cancer. Shared neoantigen vaccines are another small vaccine group and they rely on the fact that some tumor-specific neoantigens are known to be common for a subset of patients. They have the advantage of being faster and cheaper to make than totally personalized neo-antigen vaccines but have often less specific effects. In the clinical trial NCT03953235 trials immunogenicity of shared neoantigen vaccines GRT-C903 and GRT-R904 in combination with nivolumab and ipilimumab in the treatment of metastatic or advanced lung cancer, colorectal cancer, or pancreatic cancer. Another interesting study is the study NCT03552718 with a personalized neoepitope yeast-based vaccine, YE-NEO-001, for potentially curatively treated solid cancer. There are unfortunately no results yet from these clinical trials with small vaccine groups.

Unfortunately, only a few clinical results are available. These results are included in the manuscript. To take up reported results, the following changes are done:

In the heat shock protein vaccine section, the results are highlighted by changing from this:

“HSPPC-96 vaccine has also been tested in combination with bevacizumab. Interestingly, patients that only received bevacizumab without HSPPC-96 actually had the highest overall survival (10.7 months) and progression free survival (5.3 months) compared to the patients that received both bevacizumab and HSPPC-96 vaccine. (NCT01814813) This study did not include HSPPC-96 vaccine treatment alone and therefore, the overall survival for patients with HSPPC-96 vaccine alone cannot be compared to bevacizumab treatment alone. The outcome of this study potentially indicates that bevacizumab and HSPPC-96 should not be combined as a treatment.”

to this:

“HSPPC-96 vaccine has also been tested in combination with bevacizumab. Importantly, this clinical trial reported results that showed that patients that only received bevacizumab without HSPPC-96 actually had the highest overall survival (10.7 months) and progression-free survival (5.3 months) compared to the patients that received both bevacizumab and HSPPC-96 vaccine (NCT01814813). These results are important as it illustrates heat shock protein vaccines´ negative treatment outcome and shows that heat shock protein vaccines are not only connected to positive results. This study did not include HSPPC-96 vaccine treatment alone and therefore, the overall survival for patients with HSPPC-96 vaccine alone cannot be compared to bevacizumab treatment alone. The outcome of this study potentially indicates that bevacizumab and HSPPC-96 should not be combined as a treatment.”

In the dendritic cell vaccines sections, this section is changed from this:

“One study compared the efficacy of ICT-107 dendritic cells pulsed with immunogenic peptides from tumor antigen, with the efficacy of dendritic cells that had not been pulsed with antigens. Dendritic cells pulsed with antigens was observed to have median overall survival of 18.3 months, whereas dendritic cells not pulsed with antigens had the overall survival of only 16.7 months. Also, the progression free survival was higher for the group receiving dendritic cells pulsed with antigens indicating it as a potentially better treatment alternative. (NCT01280552) In another study, a patient received autologous dendritic vaccines for treatment of glioma. The vaccine was combined with poly ICLC and decitabine, but it failed to halt disease progression. (NCT02332889)”

to this:

“One study with reported results compared the efficacy of ICT-107 dendritic cells pulsed with immunogenic peptides from tumor antigen, with the efficacy of dendritic cells that had not been pulsed with antigens. The results showed that dendritic cells pulsed with antigens were observed to have a median overall survival of 18.3 months, whereas dendritic cells not pulsed with antigens had an overall survival of only 16.7 months. In a similar fashion, also the progression-free survival was higher for the dendritic cells pulsed with antigen than for the dendritic cells not pulsed with antigens. Therefore, according to the results of this trial, dendritic cells with pulsed anti-gens would be a more effective treatment alternative (NCT01280552). The study NCT02332889 also reported results but in this trial, the results were less encouraging. A patient received autologous dendritic vaccines for the treatment of relapsed pediatric high-grade glioma. The vaccine was combined with poly ICLC and decitabine, but it failed to halt disease progression (NCT02332889).”

Additionally, a limitations section is added at the end of the discussion:

“The major limitation in the neoantigen vaccine treatment is still the expensive and time-consuming development process which is required to create personalized neoantigens. Share antigens that are common for certain patient subgroups have been used to overcome this problem but with low results (NCT03953235). Peptide vaccines are often the cheapest neoantigen vaccine type to develop while mRNA vaccines are more time efficient [3, 25]. As neoantigen vaccines are often used in the treatment of late-stage cancer, patients do not necessarily have time to wait for neoantigen production. A solution for this problem would be to start with neoantigen vaccine therapy earlier in the disease progression which however would require a high number of patients to be included in the neoantigen vaccine therapies and therefore high expenses. Another limitation in the neoantigen vaccine treatment is the variable treatment outcomes that make it difficult to predict when neoantigen vaccines are reasonable to set in.”

· The authors state that from the clinical trial reports, the overall trend seems to be that neoantigen vaccination is more effective for early-stage tumors while effects are often weak for late stage tumors. However, there is no discussion on this important point, neither on the potential implications for the field (future trials, etc).

A section is added in the discussion after the statement. We discuss the difference in neoantigen vaccines effect in early-stage tumors and late-stage tumors and take up the potential future consequences of this difference:

“The results from the clinical trials show that disease-free survival time can in many cases be increased by neoantigen vaccines even in late-stage tumors, but neoantigen vaccines would do most benefit if used earlier. Unfortunately, neoantigen vaccines are often used when patients have tumor relapse or do not respond to treatments. This is because neoantigen vaccines are expensive and time-consuming to create and are, therefore, often used as a last hope rather than an early treatment option. However, if neoantigen vaccines would have been set in earlier these patients might not have relapsed or would have shown better treatment responses. Therefore, it is reasoned to set in neoantigen vaccines earlier in the treatment, for instance, to prevent relapse as done in some of the clinical trials. The problem is how to know which patients will benefit most from the neoantigen vaccine treatment as the expensive and resource-consuming treatment can often be provided only for a limited patient group. It is probable that neoantigen vaccines will be more extensively used for early-stage tumors in future clinical trials.”

· The paragraph 4 “Vaccine types” includes sub-paragraphs not always coherent with the topic of the title. For example, the sub-paragraph 4.6 “Cancer Types” on tumor indications is under the section “Vaccine types”. The same applies to the sub-paragraph 4.8 “Poly ICLC”, which could go under the “Adjuvants” paragraph. Sub-paragraph 4.12 “Vaccination” is redundant, discussing points already presented in previous sections.

All the titles are fixed to be coherent and logical. For instance: 4.8 Poly ICLC is changed to 6.1 so that it goes under 6. Adjuvants.

The repetition in the “Vaccination” section is avoided by changing the part of the section from this:

“Another vaccine type that has been able to show promising results are HSPPC -96 and -97 neoantigen vaccines. HSPPC -96 and -97 are two heat shock protein vaccines that in the similar fashion as NeoVax, are mostly trialed in the treatment of glioblastoma. In the clinical trial NCT01814813, HSPPC -96 is combined with bevaciumab. Discouragingly the results show that bevaciumab alone has better effects than when combined with heat shock proteins vaccination. Another clinical trial showed that 66% of the patients developed an immunological response after HSPPC -96 vaccination.”

to this:

“Another vaccine type that has been able to show promising results are HSPPC -96 and -97 neoantigen vaccines. HSPPC -96 and -97 are two heat shock protein vaccines that in a similar fashion as NeoVax, are mostly trialed in the treatment of glioblastoma. In the clinical trial NCT03018288, HSPPC -96 is combined with temozolomide and pembrolizumab in the treatment of glioblastoma after surgery while in another trial, NCT00293423, HSPPC-96 was trialed as monotherapy after surgical resection. This trial showed that 66% of the patients developed an immunological response after HSPPC-96 vaccination (NCT00293423).”

· Please, revise some typos/ grammatically incorrect sentences, such as:

o Paragraph 4.2: “HSPPC-96 vaccine has for instance in combination with temozolomide used in the treatment of glioblastoma”

in the conclusion paragraph: “In future, neoantigen vaccines are likely to develop further and become substantial help as a cancer treatment but further understanding of tumors´ immunosuppressive milieu and immune escape mechanisms are.” Are what? Please, complete the sentence.

The sentences above are fixed and additional English modification is done throughout the manuscript.

· The graphs in figure 1 and 2 should include y axes label. The use of the same color code for the same vaccine category shown across figures would be preferable (Fig1A and 2A).

Figures have been labeled regarding the y-axis and the same color code is used for the same vaccine category throughout the figures. Please see the figures in the reviewed version of the manuscript.